# In Vivo Evaluation of the Effects of B-Doped Strontium Apatite Nanoparticles Produced by Hydrothermal Method on Bone Repair

**DOI:** 10.3390/jfb13030110

**Published:** 2022-07-31

**Authors:** Faruk Oztekin, Turan Gurgenc, Serkan Dundar, Ibrahim Hanifi Ozercan, Tuba Talo Yildirim, Mehmet Eskibaglar, Erhan Cahit Ozcan, Cevher Kursat Macit

**Affiliations:** 1Department of Endodontics, Faculty of Dentistry, Firat University, Elazig 23100, Turkey; meskibaglar@firat.edu.tr; 2Faculty of Technology, Firat University, Elazig 23100, Turkey; tgurgenc@firat.edu.tr; 3Department of Periodontology, Faculty of Dentistry, Firat University, Elazig 23100, Turkey; sdundar@firat.edu.tr (S.D.); taloyildirim@firat.edu.tr (T.T.Y.); 4Department of Pathology, Faculty of Medicine, Firat University, Elazig 23100, Turkey; ozercanih@firat.edu.tr; 5Department of Esthetic, Plastic and Reconstructive Surgery, Faculty of Medicine, Firat University, Elazig 23100, Turkey; ecozcan@firat.edu.tr; 6Faculty of Engineering, Firat University, Elazig 23100, Turkey; macitkursatcevher@gmail.com

**Keywords:** endodontic surgery, bone repair, osteoblast, strontium apatite, nanomaterial

## Abstract

In the present study, the structural, morphological, and in vivo biocompatibility of un-doped and boron (B)-doped strontium apatite (SrAp) nanoparticles were investigated. Biomaterials were fabricated using the hydrothermal process. The structural and morphological characterizations of the fabricated nanoparticles were performed by XRD, FT-IR, FE-SEM, and EDX. Their biocompatibility was investigated by placing them in defects in rat tibiae in vivo. The un-doped and B-doped SrAp nanoparticles were successfully fabricated. The produced nanoparticles were in the shape of nano-rods, and the dimensions of the nano-rods decreased as the B ratio increased. It was observed that the structural and morphological properties of strontium apatite nanoparticles were affected by the contribution of B. A stoichiometric Sr/P ratio of 1.67 was reached in the 5% B-doped sample (1.68). The average crystallite sizes were 34.94 nm, 39.70 nm, 44.93 nm, and 48.23 nm in un-doped, 1% B-doped, 5% B-doped, and 10% B-doped samples, respectively. The results of the in vivo experiment revealed that the new bone formation and osteoblast density were higher in the groups with SrAp nanoparticles doped with different concentrations of B than in the control group, in which the open defects were untreated. It was observed that this biocompatibility and the new bone formation were especially elevated in the B groups, which added high levels of strontium were added. The osteoblast density was higher in the group in which the strontium element was placed in the opened bone defect compared with the control group. However, although new bone formation was slightly higher in the strontium group than in the control group, the difference was not statistically significant. Furthermore, the strontium group had the highest amount of fibrotic tissue formation. The produced nanoparticles can be used in dental and orthopedic applications as biomaterials.

## 1. Introduction

Apical periodontitis (AP) is a pulp-derived inflammatory disease that affects the tissues around the root apex. Radiographically, AP is seen as a radiolucent periapical lesion associated with the root tip. The first treatment option for teeth with AP is root canal treatment. When AP remains in a root-filled tooth, it appears as a radiolucent periapical lesion. Various parameters are used to evaluate the success of root canal treatment outcomes, including complete resolution of periapical pathology and clinical symptoms. It is thought that periapical problems can be best resolved with root canal treatment. The success of this treatment varies between 48% and 98%. If the root canal treatment fails, the reason for this failure should be understood before the next step of the treatment [1,2,3,4]. Root canal treatment may cause residual infections in the canal, resistant intracanal infections, extraradicular infections, cysts, foreign body reactions, or coronal leakage, and root canal treatment may fail depending on these complications. In cases in which non-surgical treatment is not possible in the maxillofacial region, periapical surgery is the only treatment option. The reported success rates for periapical surgery range from 44 to 95%. Generally, apical root tip resection is recommended if orthograde root canal treatment procedures fail to remove the reservoir of persistent microorganisms [4].

A periapical lesion is known as bone resorption at the apex of a tooth and appears as a radiolucent area in the periapical region [5]. Microorganisms that reach the periapical tissues from an infected root canal cause destruction of the periapical tissues and periapical lesion formation [6,7]. Periapical lesions may be more prone to develop in people with systemic health problems [8]. The presence of periapical lesions is higher in diabetics [9]. In addition, its incidence is higher in patients with coronary artery disease [10] and postmenopausal osteoarthritis [11].

Recently, applications for surgical endodontic treatments have increased in order to keep the tooth in the mouth. Endodontic surgery is indicated for teeth that do not heal with non-surgical endodontic treatments. The aim of endodontic surgery is to eliminate the source of infection and provide an optimal environment for periapical tissue and bone healing [12].

Placing a retrograde filling on the resected root tip in order to isolate the root canal from the periapical tissues in endodontic surgery is one of the most important steps in apical surgery. Recent studies have indicated that retrograde filling is essential for the success of apical surgical procedures [13,14].

Numerous materials, such as Cavit, super ethoxy benzoic acid, amalgam, intermediate restorative material (IRM), glass ionomer cement, compomer, and composite, have been used for root tip filling. In addition, recently, a calcium silicate-based mineral trioxide aggregate (MTA) has been used [15]. Retrograde fillings used to seal the root tip are expected to be biocompatible, prevent microleakage, and induce new bone formation.

The most well-known apatite is hydroxyapatite (HA), which contains the element Ca and is often used as a synthetic biomaterial; it is expressed by the formula Ca_10_(PO_4_)_6_OH_2_. The most important reasons why HA is often preferred as a biomaterial are its chemical properties, which are similar to those of bones and teeth; its excellent bioactivity, biocompatibility, and biodegradability; its non-toxic and non-inflammatory nature; and proven osteoconductive and osteoinductive potential [16,17,18].

Although many studies have been conducted on HA over the years, interest in Strontium (Sr) has increased in recent years due to its Ca-like properties. Almost all the Sr in the human body is found in the human skeleton. Sr is a biocompatible, non-radioactive element that reduces bone loss, strengthens the bone, improves the growth of osteocytes, has the ability to inhibit the resorption of osteoclasts, and improves bone formation by inhibiting in vitro and in vivo activity. In addition, Sr has a form that increases the proliferation of osteoblasts, supports new bone formation and osseointegration, and prevents bone resorption. It is an additive demonstrated to improve the bioavailability and bone induction of hydroxyapatite. Studies have shown that replacing the calcium ion with a strontium ion in hydroxyapatite changes its definition and improves the mechanics of crystal growth. For these reasons, Sr has become a highly preferred element in biomaterials used in bone-related treatments in recent years [19,20,21,22,23,24,25,26].

By doping elements such as Mg, Zn, Ag, Bi, Cu, Cr, Li, Co, Se, Cd, Ta, F, Eu, B, and Ni to apatite, their osteogenic, antibacterial, and biological properties can be adjusted, and biocompatible materials with different properties can thus be formed [27,28]. Recently, trace amounts of these elements in natural bone have been the focus of attention. Studies on the use of one of these elements, B, in bone tissue engineering have attracted a lot of attention. Along with its importance for osteogenic differentiation, B is a very important nutrient source for organisms that are beneficial in bone formation and bone preservation [28,29].

Nanotechnology is a very promising technology that offers customized solutions to improve the properties of biomaterials due to its low density, small size, high surface areas, and high surface/volume ratio [30]. Apatite nanoparticles can be produced by various processes, such as hydrothermal, mechanochemical precipitation, sonochemical, combustion, and sol–gel processes. The hydrothermal process has several advantages, such as simplicity, low cost, an almost homogeneous size distribution, and good crystallinity [27,31].

Furukawa produced and characterized un-doped and carbonate or silicate-substituted strontium apatite nanoparticles through the alkaline hydrolysis method [32]. In another study, Ag-doped SrAP nanoparticles were synthesized by the hydrothermal process, and their dielectric properties were investigated [27]. Slimen et al. produced sodium-, potassium-, and carbonate-doped strontium hydroxyfluorapatite and investigated their structural and thermal properties [33]. Studies on B-doped hydroxyapatite have been carried out by many researchers [28,29,34,35,36].

In the literature, there are limited studies on the biocompatibility of doped and un-doped strontium apatite (SrAp) [37,38]. In addition, studies on the in vivo biocompatibility of B-doped apatite nanoparticles are limited [28,39]. Therefore, this study produced B-doped strontium apatite nanoparticles for the first time (to the best of our knowledge); their structural characterization was performed and their in vivo biocompatibility was investigated. The aim of our study was to examine the effects of nanotechnology-produced strontium and strontium doped with different amounts of B on bone repair and biocompatibility in rats.

## 2. Material and Method

### 2.1. Fabrication of Nanoparticles

The general formula of apatite is X_10_(YO_4_)_6_Z_2_. In this formula, X represents cations such as Na^+^, Ba^2+^, Ca^2+^, Sr^2+^, and Mn^2+^, and Y represents cations such as B^3+^, V^5+^, P^5+^, As^5+^, and Si^4+^. Z represents anions such as OH^−^, F^−^, and Cl^−^. Since the X/Y ratio is 1.67 in this formula, the Sr and P ratios were chosen according to this ratio in our study. Solution A was made by dissolving 0.4 M Sigma-Aldrich (St. Louis, MO, USA) brand strontium nitrate (Sr(NO_3_)_2_) in 30 mL of deionized water. Solution B was made by dissolving 0.24 M Fluka Analytical (Munich, Germany) brand diammonium hydrogen phosphate (H_9_N_2_O_4_P) in 30 mL of distilled water in a separate baker. The solutions were stirred for 30 min at room temperature with a magnetic stirrer. Then, while stirring, solution B was dropped into solution A slowly, and the resulting mixture was stirred for 30 min. The ammonia solution was added dropwise to the mixed solution, and the pH was adjusted to 10. The mixed solution was stirred for 10 min at room temperature before being ultrasonicated for 30 min. All the above processes were performed at room temperature. The resulting fluid solution was poured into a Teflon-lined autoclave and placed in a Fytronix (Elazig, Turkey) brand hydrothermal reactor. The hydrothermal process was performed at 200 °C for 6 h. The solution was filtered after slow cooling to room temperature, and the precipitates were cleaned with alcohol and then rinsed with deionized water a few times before being dried at 80 °C for 2 h. The produced strontium apatite particles were designated SrAp after being pulverized in a mortar. To produce B-doped samples, 1 mol.%, 5 mol.%, and 10 mol.% Merck (Darmstadt, Germany) brand boric acid (H_3_BO_3_) was added to solution B, and all the above processes were repeated. Samples with B additives were named SrAp-1B, SrAp-5B, and SrAp-10B for 1 mol.%, 5 mol.%, and 10 mol.% B-doping.

### 2.2. Characterization of Nanoparticles

The XRD characterizations were made with a PANalytical Empyrean (Malvern, United Kingdom) brand diffractometer. The XRD process was performed under CuKα (λ = 1.5406Å) radiation with a scan step of 0.0262606° under 45 mA, a 45 kV current, and a scan range of 2θ = 20–80°. The lattice parameters (a = b and c), volumes of unit cells (V), crystallite size (D), and dislocation density (η) of the fabricated SrAp nanoparticles were calculated according to the relationships given by previously published studies [40,41]. The average crystallite sizes (D_ave_.) were calculated by averaging the D values calculated from the (002) and (300) diffraction planes. The dislocation intensity was also calculated using this value. The FT-IR characterizations were made using a Bruker (Billerica, MA, USA) Vertex 70v FT-IR spectrometer. The characterizations were performed with a 4000–525 cm^−^^1^ scanning range. The morphologies of fabricated nanoparticles were characterized by a Zeiss Crossbeam 540 (Oberkochen, Germany) brand FE-SEM device.

### 2.3. Animals and Study Design

This study was evaluated by the Firat University Animal Experiments Ethics Committee, and the study design was approved by decision 2021/16. All recommendations of the World Medical Association Declaration of Helsinki for the protection of laboratory test animals were followed. Statistical power analysis was used to calculate that there should be at least 8 rats in each group. In this study, 11 rats were used in each group, considering the risk of death of the animals during or after the surgical procedure. Fifty-five rats were used in our study. Bone defects 2.5 mm in diameter and 4 mm in depth were opened on the right tibia of the rats, and then the rats were divided into 5 groups (n = 11) randomly. Group 1 (n = 11) was the control group, and no procedure was applied except for creating a defect. The following were the experimental groups: group 2, strontium (n = 11); group 3, strontium + 1% boron (n = 11); group 4, strontium + 5% boron (n = 11); and group 5, strontium + 10% boron (n = 11) (Figure 1). The animals were killed after the experiments (at the end of the 8th week of the experiment). Bone healing, inflammation, and fibrosis were examined in the pathology laboratory by the hematoxylin staining method.

### 2.4. Surgical Procedures

Surgical procedures were performed under general anesthesia (ketamine (87 mg/kg; Francotar^®^, Virbac do Brasil Indústria e Comércio, Roseira, Brazil) and xylazine (13 mg/kg; Rompum^®^, Bayer S.A., Leverkusen, Germany) in sterile conditions. After full thickness dissection by taking bone contact with the scalpel over the tibial ridge, the metaphyseal part of the tibia bone was reached. Cylindrical defects 4 mm in height and 2.5 mm in diameter were created using a trephine burr (3i Implant Innovations Inc., Palm Beach Gardens, FL, USA) on the tibia under serum cooling. The produced biomaterials were placed in these bone defects with moderate pressure by a specialist in order not to disturb the standardization. Following the surgical procedure, the surgical area was closed with 3-0 Ethicon (Edinburgh, Scotland) brand Ethilon nylon dissolvable sutures. Antibiotics (penicillin 50 mg/kg, Amoxlav, Provet, Turkey) and pain relievers (tramadol hydrochloride 0.1 mg/kg, Tramadolor, Turkey) were administered for the prevention of infection and pain control after surgery. At the end of the study period, all rats were sacrificed. The biomaterials placed in the right tibia bones were decalcified, and histological bone tissue, fibrosis, and osteoblast analyses were performed. During the experimental protocol, 2 to 3 rats died in each group, so the study was continued with 8 rats in each group to ensure standardization.

### 2.5. Histopathological Procedures

Histomorphometric analyses were performed on the bone area where the defect was created and the biomaterial placed as well as the bone tissue around this area. Tibias were kept in 10% formaldehyde (Merck 103999, Darmstadt, Germany) for 72 h and demineralized in 10% formic acid (Merck, Darmstadt, Germany). After demineralization, all tibia bones were dried, embedded in paraffin, and prepared for histological sectioning. This was performed by hematoxylin–eosin staining for microscopic analysis. The bone was examined in 6 µm thick sections under a light microscope. Osteoblasts were scored as follows: no osteoblast cells = 0, slight appearance of osteoblasts = 1, sparse osteoblasts = 2, and dense osteoblasts = 3. New bone formation was scored as follows: absent = 0, slightly visible bone formation = 1, moderately visible bone formation = 2, and dense visible bone formation = 3. Fibrotic tissue was scored as follows: no fibrotic tissue = 0, superficial or focal fibrotic tissue = 1, superficial extensive or deep local fibrotic tissue = 2, and deep and extensive fibrotic tissue = 3. All histology images taken from the samples were recorded with a digital camera connected to a light microscope. Histomorphometric analyses were performed using the Olympus (Tokyo, Japan) DP71 software imaging system [42].

### 2.6. Statistical Analysis

Statistical analysis of the data was made in the IBM (Endicott, New York, NY, USA) SPSS 22 statistical package program. The Shapiro–Wilk test was used to examine whether the data showed a normal distribution. Descriptive statistics of the data are given as medians (min–max) for quantitative variables that did not show a normal distribution. For non-normally distributed quantitative data, the Kruskal–Wallis H test was used to compare more than two independent groups, and the Dunn–Bonferroni post hoc test was used for pairwise comparisons. The statistical significance level was α = 0.05.

## 3. Results

### 3.1. Structural and Morphological Characterizations of Fabricated Nanoparticles

The FT-IR spectra of the fabricated un-doped and B-doped SrAp nanoparticles are shown in Figure 2. Intense peaks were observed at wavelengths of about 900–1100 cm^−1^ in the un-doped and B-doped samples. In addition, intense peaks were observed at wavelengths of about 550–600 cm^−1^, although the intensities of these peaks were lower than those of the abovementioned peaks. The positions and intensities of the peaks were affected by B-doping. 

The XRD patterns of fabricated pure and B-doped SrAp particles are shown in Figure 3. The 2θ values of the diffraction planes are given in Appendix A. The pure SrAp matched well with JCPDS card no. 33-1348. The characteristic peaks of un-doped SrAp nanoparticles were observed at 2θ angles of 20.99°, 24.46°, 26.66°, 27.90°, 30.58°, 31.70°, 38.35°, 41.55°, 44.65°, 46.04°, 46.96°, 48.51°, and 49.95°. As the B ratio increased in the strontium apatite particles, the diffraction peak values shifted to higher 2θ values, and the peak intensities increased. The intensities and 2θ angles of the B-doped SrAp nanoparticles varied, but they perfectly matched those of JCPDS card no. 33-1348. Secondary phases or impurities were not found in un-doped and B-doped SrAp nanoparticles. The absence of secondary phases and impurities in the XRD diffraction peaks proves that the un-doped and B-doped SrAp nanoparticles were successfully produced by the hydrothermal process and had high purities. The lattice dimensions, volumes of unit cells, and crystallite size of the fabricated nanoparticles calculated from the XRD results are given in Table 1. B-doping affected all these values. The lattice dimensions and volume of the unit cells of the B-doped samples were lower than those of the un-doped sample. In addition, these values decreased as the B ratio increased. The D values of the fabricated particles were in the nanoscale range, and they increased as the B-doping ratio increased. The dislocation density decreased as the B-doping ratio increased.

Figure 4 shows the FE-SEM images of fabricated pure and B-doped SrAp particles. All fabricated SrAp particles were nano-sized and nano-rod shaped. The lengths of the nano-rods decreased as the B-doping ratio increased. The EDX analysis graphs of the fabricated nanoparticles are shown in Figure 5. Table 2 shows the EDX analysis results. The boron ratios could not be determined exactly by EDX analysis due to the low atomic number, and therefore, the ratios were high. However, the detection of the presence of B in B-doped samples provides information about the successful doping of the additive. The Sr/P ratios of SrAp, SrAp-1B, SrAp-5B, and SrAp-10B samples were calculated to be 1.76, 1.76, 1.68, and 1.58, respectively.

### 3.2. In Vivo Biocompatibility Characterizations

In terms of osteoblast level and new bone formation, the statistical results of our study showed that there were statistically significant differences between the control group and the strontium + 5% boron group and between the control group and the strontium + 10% boron group. In terms of fibrotic tissue formation, statistically significant differences were found only between the control group and the strontium group and between the strontium + 10% boron group and the strontium group. It was found that there were no statistically significant differences between the other groups for all three parameters. It was observed that the group with the highest osteoblast level and new bone formation was strontium + 10% boron, and the strontium group had the highest fibrotic tissue formation (Table 3 and Figure 6).

In the histological analysis of our study, changes were seen in the bone defects created in the rat tibia in which the biomaterials were placed. The level of osteoblast cells, new bone formation, and fibrotic tissue formation in the groups are shown (Figure 7). The results of our study suggest that adding boron to strontium yields a more biocompatible material and positively affects new bone formation.

## 4. Discussion

The important signals of the structure of apatite can be summarized as follows. The peaks observed from about 552 cm^−1^ to 593 cm^−1^ correspond to the bending vibration of the PO_4_^3−^ functional group [21,43]. The small band at about 700 cm^−1^ is ascribed to OH^-^ ions [44]. The low intense peak at about 856 cm^−1^ in the SrAp and B-doped SrAp particles is ascribed to the HPO_4_^2−^ phase, and its intensity decreased with an increasing B-doping ratio [45,46]. The peak at about 923 cm^−1^ is attributed to the bending vibration of PO_4_^3−^ [47,48]. The peaks seen from about 994 to 1060 cm^−1^ in SrAp particles are attributed to the stretching vibration of the PO_4_^3−^ functional group [21,43]. The small band at about 1452 cm^−1^ is ascribed to CO_3_^2−^ ions [43,45]. In addition, the 1574 cm^−1^ and 3510 cm^−1^ peaks are attributable to the bending and stretching of H_2_O molecules [44]. The peak intensities and positions of the phosphate peaks changed with an increase in B. There were also changes in the intensities of the OH^-^-dependent peaks [28,49]. These results show that the addition of B was carried out successfully. The FT-IR analysis results are in good agreement with recent studies [21,28,43,49]. 

The intense peaks observed in the XRD analysis of un-doped sample are ascribed to the (200), (002), (102), (210), (211), (300), (222), (312), (213), (321), and (402) planes, respectively [27]. It is thought that the increase in the intensity of the peaks and their shift to higher 2θ angles because of the B-doping in the XRD peaks are due to the lower ionic radius of B^3+^ (0.23 Å) compared with that of Sr^2+^ (1.13 Å) [28,43,50,51,52]. No secondary phases or impurities in the nanoparticles indicate that B ions are included in the SrAp lattice to substitute Sr^2+^ ions and replace Sr^2+^ ions without changing the structure of the host lattice [27,28,53,54]. The XRD results of fabricated nanoparticles are compatible with the studies in the literature [21,43,55,56,57]. It is thought that the decrease in the lattice dimensions and the volume of unit cells are due to the lower ionic radius of B^3+^ compared with that of Sr^2+^ [27,50,53,58,59]. The doping of additives with low ionic radii to hydroxyapatite decreases the lattice parameters, while additives with high ionic radii increase these parameters [27,50]. In the literature, the average values of the lattice parameters of doped and un-doped strontium apatite are a = b = 9.775 Å and c = 7.269 Å [27,43,56,57,60,61,62]. In our study, these values were 9.738 Å and 7.254 Å for a = b and c, respectively. These values were found to be compatible with those in the literature, and the crystal parameters of the B-doped samples were somewhat lower than the average values in the literature. The average volume of unit cells in the literature is 604.428 Å^3^. In our study, this value was calculated as 595.682 Å^3^ and is in good agreement with studies in the literature [27,43,56,57,60,61,62]. The crystallite sizes vary between 20.5 and 105.6 nm in previous studies on strontium apatite [27,56,57,60]. The fact that the values in our study were in this range proves that values compatible with those in the literature were obtained. The nano rod-shaped SrAp particles seen in the FE-SEM images were also obtained in previous studies [43,63]. The Sr/P ratios of all fabricated nanoparticles were close to the stoichiometric ratio of apatite (1.67) [39]. Structural and morphological analyses showed that the un-doped and B-doped SrAp nanoparticles can be successfully produced by the hydrothermal process.

Thus far, many different techniques and agents have been used in the repair of bone defects [64]. One of these agents is strontium. Strontium provides mineralization without damaging the bone tissue, increases the replication of preosteoblasts, increases the alkaline phosphatase activity, ensures osteoblast differentiation, decreases osteoclastic markers, and briefly shows both antiresorptive and osteoblastic activities in vivo. 

In addition, since strontium is a frequently preferred biomaterial in recent studies on bone tissue, it was used as a bone defect repair material in our study [65,66]. In another study, strontium-doped nano-hydroxyapatite in sheep bone defects provided bone formation from the periphery to the center of the defect. It was reported that it provided a small inflammatory infiltrate and a large amount of biomaterial to the peripheral bone matrix, as well as loose connective tissue in the central part of the defect [67]. The net effect of topically applied strontium has attracted the attention of many researchers. A group of researchers investigated the effect of strontium on fetal mouse calvarial cells upon incorporation into bioactive glass and showed that the presence of strontium (5% by weight) enhanced osteoblast differentiation [68]. The findings of our study showed parallel results to this study. In another study, hydroxyapatite was placed in one of the tibias with bilateral defects and observed for 20, 30, and 45 days and was found to be biocompatible in direct bone contact in the bone neoformation process, but compared with the control group, the material did not accelerate this process in the treated group, and no differences were observed between two analyzed groups. It was reported that it did not induce an inflammatory reaction during this period [69]. In our study, we concluded that there was an acceleration in new bone formation when strontium apatite was used instead of hydroxyapatite. However, other studies have added boron into the repair material for the repair of bone defects. In these studies, it has been shown that boron has positive effects on bone healing [20,25]. Boron (B) modulates the osteogenic effects of human bone marrow-derived mesenchymal stem cells [70,71] and mineralized tissue-related proteins as well as the adhesion and proliferation potential of osteoblasts [72]. In various studies using these elements separately, it has been reported that strontium and boron ions have positive effects on bone formation [64,65,66,73]. In the literature, there are no studies evaluating the effect of boron added in different concentrations to the strontium apatite used in the repair of bone defects on bone defect healing. In our study, the effects of 1%, 5%, and 10% boron added to strontium on the healing of bone defects were investigated. The findings of our study showed that the new bone formation and osteoblast amount were highest in the strontium + 10% boron group. In the histopathological analysis, it was observed that, compared with the control group, strontium placed in the bone cavity and boron added to the strontium had positive effects on new bone formation and osteoblast levels. The results of our study are in line with those of other studies in the literature [64,65,74,75,76]. In addition, different concentrations of boron were used in the repair/regeneration of bone defects in our study. In our study, it was observed that osteogenic activity was dependent on the dose of boron used. In this context, the findings of our study overlap with a similar study [74].

One of the limitations of our study is that in vitro evaluation could not be performed due to the lack of expert researchers and equipment at our institution. However, as a result of interviews with different universities, in vitro evaluations of these studies will be made in future studies. Another limitation of the study is the limited number of experimental animals used. Eight experimental animals were used in this study. A higher number may result in better results. However, for the protection of laboratory experimental animals, it was calculated by statistical power analysis that at least eight rats should be included in each group, in accordance with the recommendations of the Declaration of Helsinki World Medical Association. In our country, it is recommended to use the minimum number of animals in accordance with the ethical rules of animal experiments.

## 5. Conclusions

Boron-doped strontium apatite nanoparticles were successfully produced using the hydrothermal method. It was determined that the nanoparticles were in the shape of nano-rods, and the lengths of the nano-rods decreased as the B-doping increased. According to the limited results of this study, it can be put forward that local boron and strontium application may be used successfully as retrograde fillers during apical surgery. In addition, it can be considered that local boron application may be a more advantageous application in terms of bone healing. There is a need for further studies examining the relationship between local boron and strontium application and the bone tissue, and different doses and administration methods should be investigated. The fabricated un-doped and B-doped strontium apatite nanoparticles can be used in orthopedic and dental applications as biomaterials.

## Figures and Tables

**Figure 1 jfb-13-00110-f001:**
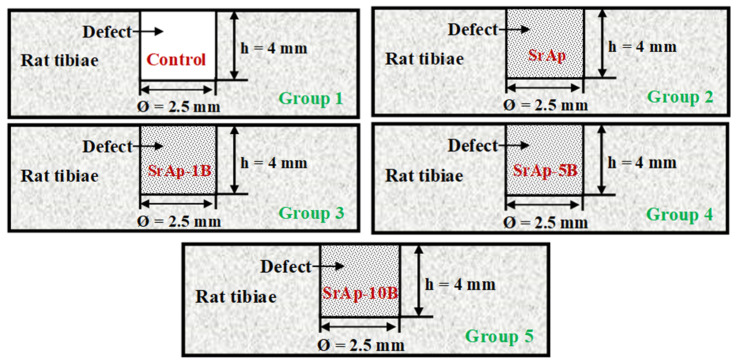
Experimental design and groups.

**Figure 2 jfb-13-00110-f002:**
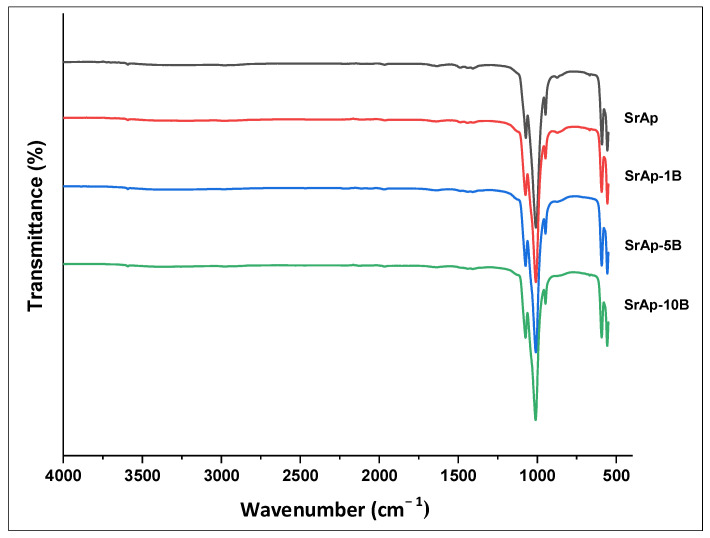
FT-IR spectra of the fabricated nanoparticles.

**Figure 3 jfb-13-00110-f003:**
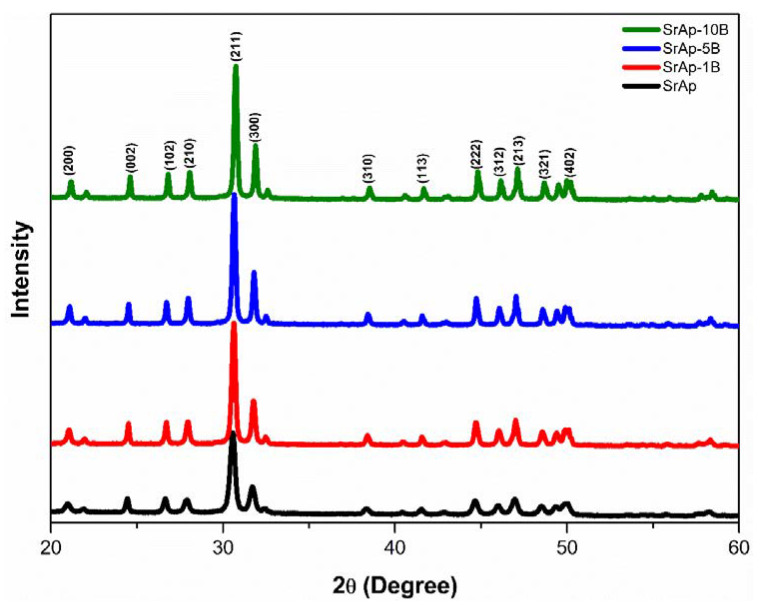
XRD results of fabricated nanoparticles.

**Figure 4 jfb-13-00110-f004:**
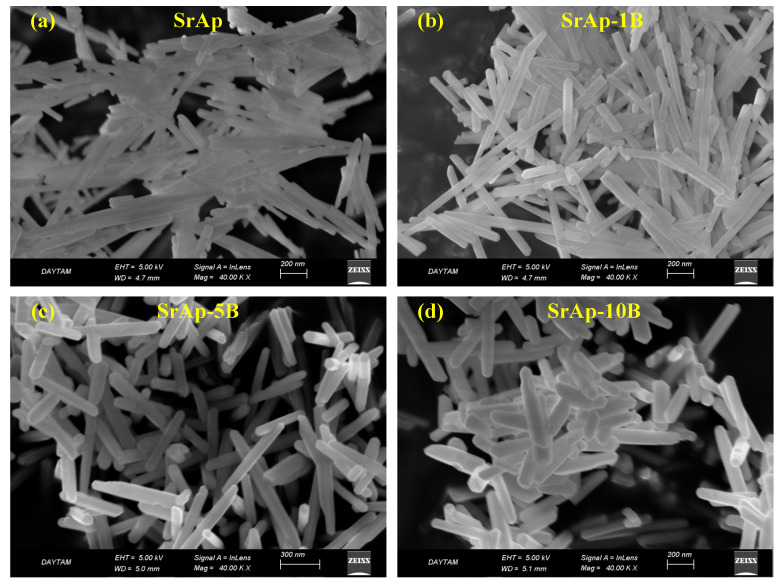
FE-SEM images of fabricated nanoparticles (**a**) SrAp, (**b**) SrAp-1B, (**c**) SrAp-5B and (**d**) SrAp-10B.

**Figure 5 jfb-13-00110-f005:**
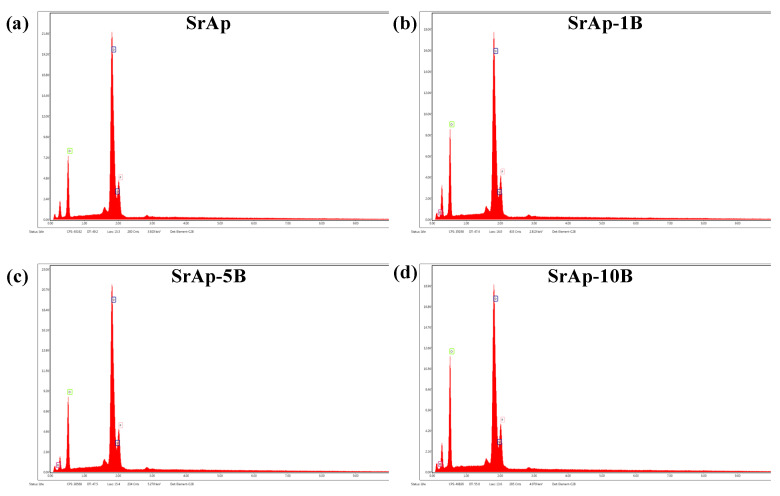
EDX graphics of fabricated nanoparticles (**a**) SrAp, (**b**) SrAp-1B, (**c**) SrAp-5B and (**d**) SrAp-10B.

**Figure 6 jfb-13-00110-f006:**
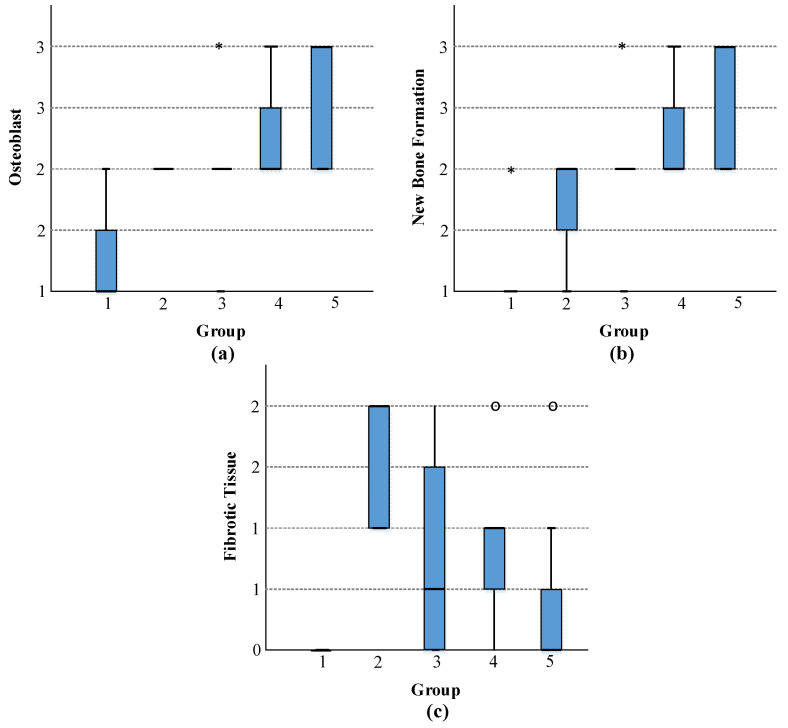
Statistical results for (**a**) osteoblasts, (**b**) new bone formation, and (**c**) fibrotic tissue. (o = Outlier value and * = Extreme outlier value).

**Figure 7 jfb-13-00110-f007:**
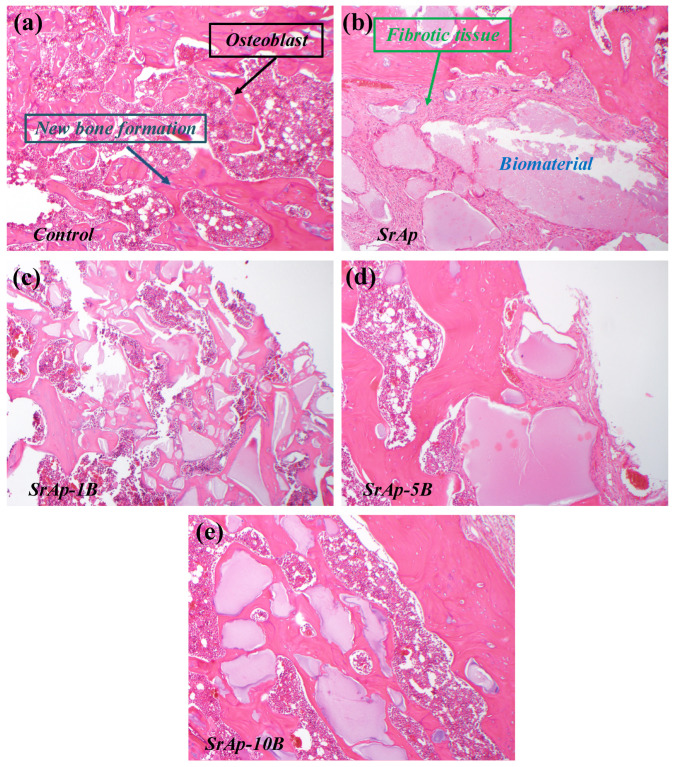
The level of osteoblast cells, new bone formation, and fibrotic tissue formation in the different groups (**a**) Control, (**b**) SrAp, (**c**) SrAp-1B, (**d**) SrAp-5B and (**e**) SrAp-10B.

**Table 1 jfb-13-00110-t001:** The a, b, c, V, and D values of fabricated SrAp particles from XRD results.

Sample	a = b (Å)	c (Å)	V (Å^3^)	D_002_ (nm)	D_300_ (nm)	D_ave_. (nm)	η (nm^−2^)
SrAp	9.7701	7.2726	601.176	51.67	18.21	34.94	8.191 × 10^−4^
SrAp-1B	9.7461	7.2580	597.027	49.01	30.38	39.70	6.345 × 10^−4^
SrAp-5B	9.7282	7.2492	594.123	52.61	37.24	44.93	4.954 × 10^−4^
SrAp-10B	9.7074	7.2347	590.402	56.34	40.11	48.23	4.299 × 10^−4^

**Table 2 jfb-13-00110-t002:** EDX analysis results of fabricated nanoparticles (at.%).

Sample	Sr	P	O	B	Sr/P
SrAp	25.19	14.32	60.49	-	1.76
SrAp-1B	11.50	6.55	37.79	44.16	1.76
SrAp-5B	17.57	10.47	48.83	23.13	1.68
SrAp-10B	11.22	7.11	45.56	36.11	1.58

**Table 3 jfb-13-00110-t003:** Intergroup comparison of variables.

	Osteoblast Median (Min–Max) (n = 8)	New Bone Formation Median (Min–Max) (n = 8)	Fibrotic Tissue Median (Min–Max) (n = 8)
**Control (1)**	1 (1–2)	1 (1–2)	0 (0–0)
**Strontium (2)**	2 (2–2)	2 (1–2)	2 (1–2)
**Strontium + 1% Boron (3)**	2 (1–3)	2 (1–3)	0.5 (0–2)
**Strontium + 5% Boron (4)**	2 (2–3)	2 (2–3)	1 (0–2)
**Strontium + 10% Boron (5)**	3 (2–3)	3 (2–3)	0 (0–2)
** *p* **	<0.001	<0.001	<0.001
***p* ***	1–4: *p* = 0.014 1–5: *p* < 0.001 Others: *p* > 0.05	1–4: *p* = 0.007 1–5: *p* < 0.001 Others: *p* > 0.05	1–2: *p* = 0.001 2–5: *p* = 0.020 Others: *p* > 0.05

***p*:** Kruskal–Wallis H test, ***p* *:** Dunn–Bonferroni test (pairwise).

## Data Availability

The data presented in this study are available on request from the corresponding author.

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
