# Peer review of "In Vivo Evaluation of the Effects of B-Doped Strontium Apatite Nanoparticles Produced by Hydrothermal Method on Bone Repair"

_jfb, 2022, doi:10.3390/jfb13030110_

Round 1
Reviewer 1 Report
The manuscript by OZTEKIN et. al entitled ” In vivo biocompatibility of B-doped Strontium Apatite Nanoparticles Fabricated by Hydrothermal Method: structural, morphological characterizations, and bone regeneration histological analysis in rats”, aims to assess structural, morphological and in-vivo biocompatibility of un-doped 16 and boron doped strontium apatite nanoparticles.
Evaluating the work, I would like to make a few observations:
- In the introduction in the last paragraph, reference could be made to the limited number of studies mentioned.
- In methods it needs to be left “product name, Sigma-Aldrich (St. Louis, MO, USA)” for example
- As in the previous case, state and country details must be indicated on the equipment (line 141 for example). Do this in the text as a whole.
- The basic formulas shown in lines 145, 14, 7 and 151 could only be referenced and do not need to be explained.
- For a better understanding of the in vivo study in the experimental design, a figure can be drawn up in item 2.3 and thus it will be easier to understand the groups and experiments
- In item 2.4 you need to put the name of the supplier, brand to the products used. This type of observation extends to the work as a whole.
- Do the values ​​obtained show no error?
- Need to indicate the limitations of the work
- The physicochemical characterization of the nanoparticles was carried out, as well as an in vivo study, but I thought it was important to carry out the in vitro evaluation that I did not observe in the manuscript. Need to complement.
Author Response
Respond to Reviewer
We sincerely thank the reviewer for constructive criticisms and valuable comments, which were of great help in revising the manuscript. The comments, which significantly contributed to improving the quality of the publication are quite helpful for us. Accordingly, the revised manuscript has been systematically improved with new information and additional interpretations. Please find below a detailed response to the each of the comments. And we hope the Editors and the Reviewer will be satisfied with our responses to the ‘comments’ and the revisions for the original manuscript.
Reviewer #1:
The manuscript by OZTEKIN et. al entitled ” In vivo biocompatibility of B-doped Strontium Apatite Nanoparticles Fabricated by Hydrothermal Method: structural, morphological characterizations, and bone regeneration histological analysis in rats”, aims to assess structural, morphological and in-vivo biocompatibility of un-doped 16 and boron doped strontium apatite nanoparticles.
Evaluating the work, I would like to make a few observations:
- In the introduction in the last paragraph, reference could be made to the limited number of studies mentioned.
2.In methods it needs to be left “product name, Sigma-Aldrich (St. Louis, MO, USA)” for example
- As in the previous case, state and country details must be indicated on the equipment (line 141 for example). Do this in the text as a whole.
- The basic formulas shown in lines 145, 14, 7 and 151 could only be referenced and do not need to be explained.
- For a better understanding of the in vivo study in the experimental design, a figure can be drawn up in item 2.3 and thus it will be easier to understand the groups and experiments
- In item 2.4 you need to put the name of the supplier, brand to the products used. This type of observation extends to the work as a whole.
- Do the values ​​obtained show no error?
- Need to indicate the limitations of the work.
- The physicochemical characterization of the nanoparticles was carried out, as well as an in vivo study, but I thought it was important to carry out the in vitro evaluation that I did not observe in the manuscript. Need to complement.
To Reviewer 1:
- The authors would like to thank the reviewer for his/her valuable comments on this subject. Necessary citations have been added to the part you specified in the study.
- Thank you for taking the time to spot this issue. Company information’s has been added to the relevant parts of the study.
- The authors would like to thank the reviewer for his/her suggestion. Company information’s of used devices have been added to the relevant parts of the study.
- The authors would like to thank the reviewer for his/her recommendations. The specified part was arranged as follows.
The lattice parameters (a=b and c), volumes of unit cells (V), the crystallite size (D) and dislocation density (η) of fabricated SrAp nanoparticles were calculated from relations given in literature studies [40,41]. The average crystallite sizes (Dave.) were calculated by averaging the D values calculated from the (002) and (300) diffraction planes. Dislocation intensity was also calculated using this value.
- The authors would like to thank the reviewer for his/her suggestion. The figure showing the experimental design and groups has been added to the part you suggested in the study.
- The authors would like to thank the reviewer for his/her valuable comments on this subject. Company information’s of used devices, chemicals and drugs have been added to the relevant parts of the study.
- Thank you for taking the time to spot this issue. Although it is not fully understood which values are mentioned, structural and statistical analysis values were re-examined and 25.20 was corrected to 25.19 due to rounding only in pure SrAp sample. It may appear that an error has been made in the mean crystallite sizes. The mean crystallite sizes were found by averaging the values calculated from the (002) and (300) diffraction planes, and this is stated in the material and method. No other errors were found. Our work has been analyzed very precisely. If there is a value that you think we have overlooked, you can indicate it.
- Thank you very much for your suggestions to make the study stronger. The limitations of our study were added as a separate paragraph under the discussion section.
- The authors would like to thank the reviewer for his/her suggestion on how to make our paper stronger. Since we consider the in vivo evaluation of the biomaterials produced in our study to be the first important step in terms of clinical use, we performed this study only in vivo. Due to the lack of experts and equipment for in vitro experiments in our study, we could not make the aforementioned evaluation in this process. However, we are aware of the importance of this subject and we have contacted experts from different Universities for in vitro evaluation and we plan to present this study as a separate research as soon as possible.

Reviewer 2 Report
The paper entitled “ In vivo biocompatibility of B-doped Strontium Apatite Nanoparticles Fabricated by Hydrothermal Method: structural, morphological characterizations, and bone regeneration histological analysis in rats” focuses on the effects of 1%, 5%, and 10% boron added to strontium on the healing of bone defects. The tests (XRD, FT-IR, FE-SEM, and EDX) are versatile and complementary while the results obtained are interesting and promising. The introduction refers to the aim of the study, the experimental part is consistently revealed and explained while the results are understandably submitted and sufficiently illustrated. The conclusion summarizes the aforementioned results. In my opinion, the paper will be interesting from a scientific and practical point of view. However, I would like to recommend the publication of the manuscript in this journal after fulfilling the following recommendations:
1. The title could be re-written because the text is becoming too clumsy.
2. In the abstract, more quantitative findings can be included;
3. The formulas pointed out in section 2.2 are well known and only referenced could be used.
4. How was the concentration of Sr chosen. This should be carefully explained in the text;
5. Table 1 can be moved to supplementary files;
6. The dislocation density of the doped and un-doped samples can also be calculated and compared.
7. It is not clear from the text how “Graft beds of 4 mm height and 2.5 mm width…” were created. What was the porosity of these grafts – the same for all samples or different?.
8. The long-lasting stability and the possible Sr and P release from the nanoparticles in water or SBF should also be revealed. This is important since, at certain concentrations, the presence of both elements in the organism could be harmful.
9. In Table 2, the date should be compared with specific reference values.
10. Some technical comments:
-The sentence “During the experimental protocol, 3 rats were died in each group and thus the study ended with 8 rats in each group..” should be corrected.
Author Response
Respond to Reviewer
We sincerely thank the reviewer for constructive criticisms and valuable comments, which were of great help in revising the manuscript. The comments, which significantly contributed to improving the quality of the publication are quite helpful for us. Accordingly, the revised manuscript has been systematically improved with new information and additional interpretations. Please find below a detailed response to the each of the comments. And we hope the Editors and the Reviewer will be satisfied with our responses to the ‘comments’ and the revisions for the original manuscript.
Reviewer #2:
The paper entitled “In vivo biocompatibility of B-doped Strontium Apatite Nanoparticles Fabricated by Hydrothermal Method: structural, morphological characterizations, and bone regeneration histological analysis in rats” focuses on the effects of 1%, 5%, and 10% boron added to strontium on the healing of bone defects. The tests (XRD, FT-IR, FE-SEM, and EDX) are versatile and complementary while the results obtained are interesting and promising. The introduction refers to the aim of the study, the experimental part is consistently revealed and explained while the results are understandably submitted and sufficiently illustrated. The conclusion summarizes the aforementioned results. In my opinion, the paper will be interesting from a scientific and practical point of view. However, I would like to recommend the publication of the manuscript in this journal after fulfilling the following recommendations:
- The title could be re-written because the text is becoming too clumsy.
- In the abstract, more quantitative findings can be included;
- The formulas pointed out in section 2.2 are well known and only referenced could be used.
- How was the concentration of Sr chosen. This should be carefully explained in the text;
- Table 1 can be moved to supplementary files;
- The dislocation density of the doped and un-doped samples can also be calculated and compared.
- It is not clear from the text how “Graft beds of 4 mm height and 2.5 mm width…” were created. What was the porosity of these grafts – the same for all samples or different?.
- The long-lasting stability and the possible Sr and P release from the nanoparticles in water or SBF should also be revealed. This is important since, at certain concentrations, the presence of both elements in the organism could be harmful.
- In Table 2, the date should be compared with specific reference values.
- Some technical comments:
The sentence “During the experimental protocol, 3 rats were died in each group and thus the study ended with 8 rats in each group..” should be corrected.
To Reviewer 2:
- Thank you for taking the time to spot this issue. The title of the study has been corrected as follows.
‘’In vivo evaluation of the effects of B-doped Strontium Apatite Nanoparticles Produced by Hydrothermal Method on bone repair’’
- The authors would like to thank the reviewer for his/her suggestion. The following section has been added to the abstract.
‘’The stoichiometric Sr/P ratio of 1.67 was reached in the 5% B doped sample (1.68). The crystallite sizes are 34.94 nm, 39.70 nm, 44.93 nm and 48.23 nm in un-doped, 1% B, 5% B and 10% B doped samples, respectively.’’
- The authors would like to thank the reviewer for his/her recommendations. The specified part was arranged as follows.
The lattice parameters (a=b and c), volumes of unit cells (V), the crystallite size (D) and dislocation density (η) of fabricated SrAp nanoparticles were calculated from relations given in literature studies [40,41]. The average crystallite sizes (Dave.) were calculated by averaging the D values calculated from the (002) and (300) diffraction planes. Dislocation intensity was also calculated using this value
- The authors would like to thank the reviewer for his/her valuable comments on this subject. How the Sr ratio is selected is included in the study as follows.
‘’The general formula of apatites is X10(YO4)6Z2. In this formula, X represents cations such as Na+, Ba2+, Ca2+, Sr2+ and Mn2+, and Y represents cations such as B3+, V5+, P5+, As5+ and Si4+. Z represents anions such as OH−, F−, and Cl−. Since the X/Y ratio is 1.67 in this formula, Sr and P ratios were chosen according to this ratio in our study.’’
- Thank you for taking the time to spot this issue. Table 1 has been added to the supplementary files.
- The authors would like to thank the reviewer for his/her recommendations. Dislocation densities were calculated, added to Table 2 and necessary comparisons were made.
- The authors would like to thank the reviewer for his/her recommendations to our paper stronger. How bone defects are created is explained in the text as follows. Figure 1 was added to the study for better understanding of the experimental design and groups. The porosities of the grafts were not measured. However, since the dimensions of the biomaterials used in the study were approximately the same and this part of the study was carried out by only one expert researcher, the porosities are thought to be similar.
‘’ Cylindrical defects 4 mm in height and 2.5 mm in diameter were created using a tre-phine burr (3i Implant Innovations Inc., Palm Beach Gardens, FL, USA) on the tibia under serum cooling. Produced biomaterials were placed in these bone defects with moderate pressure only by a specialist in order not to disturb the standardization.’’

Round 2
Reviewer 1 Report
All the observations pointed out were carried out
Reviewer 2 Report
The authors have carefully addressed the reviewer's recommendations and the paper may be accepted in its present form.